# BioNAS: Incorporating Bio-inspired Learning Rules to Neural Architecture Search

## Abstract

Bio-inspired neural networks have gained traction due to their adversarial robustness, energy efficiency and for being biologically plausible. While these bio-inspired networks have shown significant progress, they still fall short in terms of accuracy and are hard to scale to complex tasks. In this paper, we propose to use neural architecture search to further improve state-of-the-art bio-inspired neural networks. We achieve this thanks to BioNAS, a framework for neural architecture search designed for bio-inspired neural networks. It explores different bio-inspired neural network architectures and explores also the use of different learning rules for the layers of models being explored. The motivation for this choice comes from recent work in the field suggesting that different learning mechanisms might be used in different regions of the human brain. Using BioNAS, we get state-of-the-art bio-inspired neural network performance achieving an accuracy of 94.86 on CIFAR10, 76.48 on CIFAR-100 and 43.42 on ImageNet16-120, surpassing state-of-the-art bio-inspired neural networks. We show that a part of this improvement comes from the use of different learning rules instead of using a single learning rule for all the layers. To the best of our knowledge, BioNAS is the first neural architecture search framework that allows the exploration of using different bio-inspired learning rules for a neural network. We release BioNAS to the community and make the code available via this link (https://anonymous.4open.science/r/LR-NAS-DFE1).

Bio-inspired neural networks have gained significant traction recently. They are more biologically plausible, more robust to adversarial attacks Sanfiz & Akrout (2021a); Moraitis et al. (2022a), and have higher energy efficiency Malcolm & Casco-Rodriguez (2023). Despite their significant success, these methods are still in development and fall short in terms of accuracy and in scaling effectively for complex tasks. In this paper, we aim to use neural architecture search to improve state-of-the-art bio-inspired neural networks.

Neural Architecture Search (NAS) is a method that automatically searches for neural network architectures and effectively finds high-quality architectures. This is usually done by exploring different types of architectures and picking one that best satisfies an objective (e.g., minimizing the loss). Early work based on Evolutionary Algorithms (EAs) and Reinforcement Learning (RL) was able to find interesting architectures, however such methods take significant amount of time. AmoebaNet-B Real et al. (2019), for example, takes around 3150 GPU days. Several efficient NAS methods have been developed since then, offering a trade-off between search time and accuracy. Differentiable Architecture Search (DARTS) Liu et al. (2018) is a notable example. It has shown success in designing differentiable neural architecture search. EG-NAS Cai et al. (2024), a follow-up work, improved on top of DARTS using genetic algorithms. It has shown an ability to find an accuracy competitive with DARTS in 0.1 GPU days compared to 1 GPU day for DARTS on CIFAR10.

In this paper, we build on top of DARTS and EG-NAS to propose BioNAS, a framework for neural architecture search that explores different bio-inspired neural network architectures and learning rules. BioNAS is the first NAS framework that explores the use of different bio-inspired learning rules for different layers of the model. The motivation for this choice comes from recent work in the field suggesting that the brain uses different learning mechanisms across different layers and regions in the brain, from Hebbian Learning to more complex optimization techniques Marblestone et al. (2016). To the best of our knowledge, this is the first work to address incorporating learning rules as part of the neural architecture search. We show that interesting architectures can be discovered

within this framework surpassing state-of-the-art bio-inspired neural networks, and achieving an accuracy of 94.86 on CIFAR10, 76.48 on CIFAR-100 and  43.42  on ImageNet16-120. We show that a part of this improvement comes from the use of different learning rules instead of using a single algorithm for all the layers.

The contributions of this paper are as follows

1. We propose BioNAS, a framework for neural architecture search designed for bio-inspired neural networks.

2. To the best of our knowledge, BioNAS is the first framework that incorporates learning rules as part of the neural architecture search.

3. We show that using different bio-inspired learning rules, improves the accuracy of bio-inspired models as well as their adversarial robustness.

4. We release BioNAS to the community and make the code available via this link (https://anonymous.4open.science/r/LR-NAS-DFE1).

## 1 RELATED WORK

Backpropagation (BP) Rumelhart et al. (1986) is the most commonly used algorithm to train neural networks it relies on a forward pass where a prediction is made, and a backward pass that updates the weights according to the derivative of the loss. The backward pass relies on a weight-transposed matrix and requires symmetry. This is known as the weight transport problem. Many alternatives have been proposed to relax this requirement especially since evidence from the brain has shown that it doesn't learn this way. Lillicrap et al. (2016) has proposed Feedback Alignment (FA) in its simplest form that instead of using a transposed weight gradient matrix in the backward pass, it uses a fixed random feedback matrix that's shared across the network. Another variant called Direct Feedback Alignment has been proposed by Nøkland (2016) and uses a random feedback matrix shared across the network, however, the way the feedback synapses are wired is different: it is skipped directly to each layer separately from the output layer. Coherent learning happens even with these random feedback matrices, credit assignment occurs as a result of a partial alignment of the feedforward and backward weights. Liao et al. (2016) has shown that by using different sign concordance matrices, the performance can be improved and can approximate backpropagation. Sanfiz & Akrout (2021a) has benchmarked these different techniques and their intrinsic adversarial robustness.

On another note, Dellaferrera & Kreiman (2023) proposes "Present the Error to Perturb the Input To modulate Activity" (PEPITA), which is an algorithm that removes the backward pass and uses a second forward pass instead, to modulate the input signal based on the error. Hinton (2022) proposed the Forward-Forward algorithm that relies on a goodness metric for each layer and two forward passes. The goodness metric can theoretically be set to any metric that is high for positive data and low for negative data, such as the sum of the squared activities in a layer, this algorithm has been tested on simple tasks on shallow networks and has yet to be scaled for complex image classification tasks.

Brain-inspired plasticity rules such as Hebbian Learning have long been used in computational neuroscience as a plasticity rule implemented by certain synapses in the brain. Yet this mechanism of strengthening the synaptic weights when neurons fire together or inhibit them (Anti-Hebbian Learning) remained very simple and ineffective in learning complex tasks,  Amato et al. (2019); Lagani et al. (2022b) have trained convolutional neural networks on complex tasks such as CIFAR10 and ImageNet, however Journé et al. (2023) have shown that Hebbian Learning can scale up and perform well on these complex tasks in an unsupervised way using a Winner-Take-All mechanism. Their method relies heavily on the choice of hyperparameters and relies on an adaptive learning rate.

Adversarial attacks are malicious attacks that try to fool models to erode their predictions. They use information about the architecture, the data that it has been trained on, or simply in a black box manner. Famous adversarial attacks include One or Few Pixel attacks Su et al. (2019), PGD Madry (2017), FGSM Goodfellow et al. (2014). Wu et al. (2024) have benchmarked the adversarial robustness of architectures resulting from NAS. Few works have investigated the search for robust architectures such as Zhu et al. (2023), where they include the adversarial score in the objective func-

tion while searching, this involves extra-evaluation of the resulting architectures. We're interested in exploring NAS for bio-inspired neural networks due to their robustness to noise and attacks without needing extra evaluation, especially since backpropagation-trained neural networks are more prone to gradient-based attacks.

## 2 BACKGROUND

In this work, we consider a neural network characterized by an input vector $\mathbf{z}$, an output vector $\mathbf{y}$, and an activation function $\phi(\cdot)$. The conventional weight update in backpropagation is derived using the chain rule of calculus and is given by:

$$\delta W_i = -\eta \frac{\partial L}{\partial W_i} = -\eta \left( \frac{\partial L}{\partial \mathbf{y}} \cdot \frac{\partial \mathbf{y}}{\partial \mathbf{z}_i} \cdot \frac{\partial \mathbf{z}_i}{\partial W_i} \right), \tag{1}$$

where $\eta$ is the learning rate, $L$ is the loss function, and $\mathbf{z}_i$ represents the pre-activation at layer $i$. The gradient $\frac{\partial \mathbf{z}_i}{\partial W_i}$ is typically calculated as:

$$\frac{\partial \mathbf{z}_i}{\partial W_i} = \phi'(\mathbf{z}_i)\mathbf{y}_{i-1}^T, \tag{2}$$

where $\mathbf{y}_{i-1}$ is the output from the previous layer, and $\phi'(\mathbf{z}_i)$ is the derivative of the activation function with respect to $\mathbf{z}_i$. This update rule relies on the precise gradients provided by the subsequent layer's weight matrix, $W_{i+1}$.

### 2.1 BIO-INSPIRED LEARNING RULES

We base our work on learning rules discussed by Sanfiz & Akrout (2021a); Liao et al. (2016).

#### FEEDBACK ALIGNMENT (FA)

In layer $i$, the conventional weight update method necessitates knowledge of the subsequent layer's weight matrix, $W_{i+1}$, which is biologically implausible as it requires neurons to exchange extensive synaptic weight data. This challenge, often referred to as the weight transport problem, is addressed by substituting $W_{i+1}$ with a randomly generated synaptic weight matrix $B_{i+1}$. This adjustment enables a distinct backward pass to circumvent the weight transport issue, modifying the weight update equation to:

$$\delta W_i = -((B_{i+1}\delta\mathbf{z}_{i+1})\phi'(\mathbf{z}_i))\mathbf{y}_{i-1}, where \delta\mathbf{z}_{i+1} = \frac{\partial L}{\partial \mathbf{z}_{i+1}}. \tag{3}$$

#### UNIFORM SIGN-CONCORDANT FEEDBACKS (uSF)

This technique transmits only the sign of the forward matrix weights, assuming each synaptic weight has a unit magnitude:

$$B_i = \text{sign}(W_i) \text{ for all } i. \tag{4}$$

#### BATCHWISE RANDOM MAGNITUDE SIGN-CONCORDANT FEEDBACKS (brSF)

In this method, the magnitude of the synaptic weights $|R_i|$ for each layer is recalculated after every update, configured as:

$$B_i = |R_i|\text{sign}(W_i) \text{ for all } i. \tag{5}$$

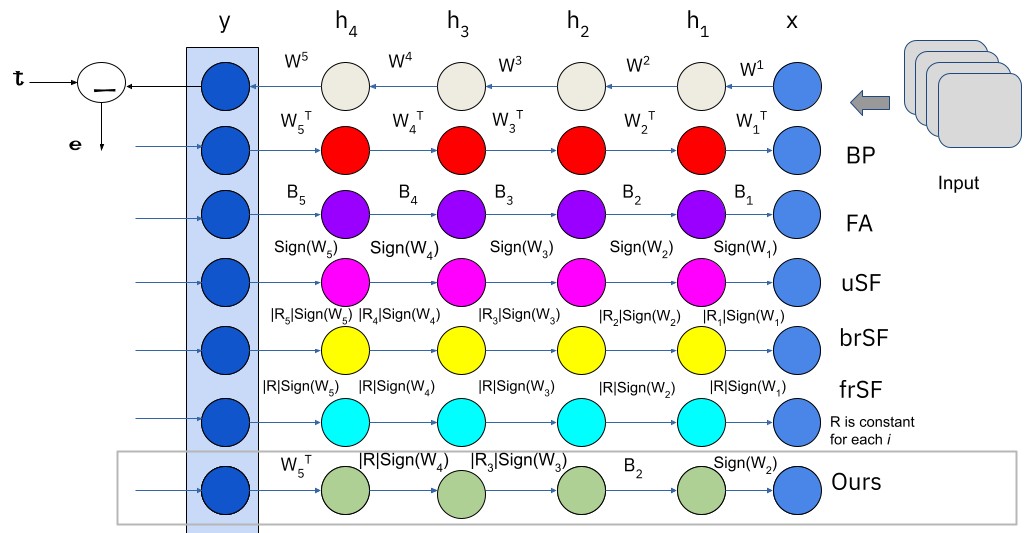

Figure 1: An illustration showing the usage of different credit assignment with BP and FA techniques.

FIXED RANDOM MAGNITUDE SIGN-CONCORDANT FEEDBACKS (FRSF)

A variant of brSF, where the magnitude of the weights $|R_i|$ remains constant $|R|$ and is set at the beginning of training:

$$B_i = |R|\text{sign}(W_i) \text{ for all } i. \tag{6}$$

## 2.2 DARTS

Differentiable Architecture Search (DARTS) Liu et al. (2018); Chu et al. (2020) is a neural architecture search (NAS) method that performs continuous relaxation of the search space to enable differentiable optimization. This technique reduces the computational cost by enabling weight sharing across candidate architectures during the search phase. In DARTS, a supernetwork composed of stacked normal and reduction cells is trained, with reduction cells positioned at certain layers to downsample feature maps and capture hierarchical representations.

Each cell in DARTS is represented as a directed acyclic graph (DAG), where the nodes correspond to feature maps, and the edges correspond to operations such as convolutions or pooling. The search process optimizes architecture parameters $\alpha$ alongside the network weights $w$ via gradient-based optimization. A softmax over the candidate operations defines the mixed operation for each edge in the DAG:

$$\bar{O}^{(i,j)}(x) = \sum_{o \in \mathcal{O}} \frac{\exp(\alpha_o^{(i,j)})}{\sum_{o' \in \mathcal{O}} \exp(\alpha_{o'}^{(i,j)})} o(x) \tag{7}$$

After the search phase, the architecture is derived by selecting the operations with the highest weights for each edge, thus producing a finalized network architecture based on learned architecture parameters. We provide more details in Appendix A.4.1.

## 2.3 EG-NAS

EG-NAS Cai et al. (2024) builds upon DARTS by integrating evolutionary strategies into the search process, specifically leveraging Covariance Matrix Adaptation Evolution Strategy (CMA-

ES) Hansen (2016); Loshchilov & Hutter (2016). EG-NAS alternates between gradient descent to update network weights and evolutionary updates to explore the architecture parameters $\alpha$. By sampling architectures from a Gaussian distribution and evolving them using CMA-ES, EG-NAS enhances the exploration of the search space and reduces the risk of converging to local optima.

The use of evolutionary strategies allows EG-NAS to scale more effectively for larger and more complex search spaces compared to DARTS, particularly when rapid exploration of architecture variations is required, more details about this algorithm are provided in Appendix A.4.2.

## 2.4 Adversarial Attacks

We evaluate the robustness of bio-inspired neural networks against the following adversarial attacks.

- **Fast Gradient Sign Method (FGSM)** Goodfellow et al. (2014): A white-box attack that perturbs the input in the direction of the gradient of the loss with respect to the input, bounded by an $L_\infty$ norm constraint.
- **Projected Gradient Descent (PGD)** Madry (2017): An iterative extension of FGSM, which applies multiple small perturbations to maximize the loss, followed by a projection back onto the $L_\infty$ constraint set.
- **Targeted Projected Gradient Descent (TPGD)** Zhang et al. (2019): A variant of PGD that uses KL-divergence loss to balance robustness and accuracy.
- **One-Pixel Attack** Su et al. (2019): An attack that manipulates only a few pixels in the input image to fool the neural network while retaining the visual integrity of the input.
- **Auto Projected Gradient Descent (APGD)** Croce & Hein (2020): A parameter-free iterative attack that automatically adjusts the step size and adapts to the model's gradients for more efficient and robust perturbation.

## 3 Methods

Our method builds upon DARTS and EG-NAS but introduces learning rules in the search space. Specifically, we propose BioNAS-DARTS and BioNAS-EG, which extend the baseline methods by incorporating a more comprehensive set of candidate operations based on bio-inspired learning rules.

We implement different variants of feedback alignment, each based on a backward bio-inspired connection. For simplicity, we do not use DFA as it requires a change in the computational graph of the neural network, which would affect other methods relying on a backward pass using PyTorch. We also do not use methods that rely on two forward passes such as Hinton (2022); Dellaferrera & Kreiman (2023). Instead, we rely on other feedback alignment techniques that can be separated into blocks and are not skip-connected.

## 3.1 Search Space

The key contribution of our work lies in the design of the search space, where we expand the set of candidate operations to include operations guided by bio-inspired learning rules. We refer to these learning rules include Feedback Alignment (FA), Uniform Sign-concordant Feedback (uSF), Batchwise Random Magnitude Sign-concordant Feedback (brSF), and Fixed Random Magnitude Sign-concordant Feedback (frSF). The operations in our search space are as follows:

**Candidate Operations:**

- $3 \times 3$ separable convolutions (in 4 modes: FA, uSF, brSF, frSF)
- $5 \times 5$ separable convolutions (in 4 modes: FA, uSF, brSF, frSF)
- $3 \times 3$ dilated convolutions (in 4 modes: FA, uSF, brSF, frSF)
- $5 \times 5$ dilated convolutions (in 4 modes: FA, uSF, brSF, frSF)
- $3 \times 3$ max pooling

- $3 \times 3$ average pooling
- Skip connection (Identity if stride equals 1, or Factorized reduce if stride equals 2, with 3 modes: uSF, brSF, frSF)
- Zero (no operation, it is used for DARTS and omitted for EG-NAS like the baseline)

This enriched search space allows us to explore a wider variety of robust architectures and adapt to different learning paradigms. Figure 2 shows that for each block selected a learning rule is selected as well at the same time.

### 3.2 BIONAS-DARTS

BioNAS-DARTS builds upon DARTS by introducing the expanded search space detailed above. During the search phase, both architecture parameters $\alpha$ and network weights $w$ are optimized simultaneously. The search is conducted over a supernetwork composed of normal and reduction cells, where each edge between nodes represents a candidate operation from our enhanced operation set. The continuous relaxation method of DARTS allows us to optimize over this complex search space efficiently. Figure 3 illustrates the search strategy for BioNAS-EG.

In our framework, each operation on the edge can be selected from multiple bio-inspired modes, making the search process more diverse and capable of handling different types of data and learning tasks.

### 3.3 BIONAS-EG

BioNAS-EG utilizes the faster, more scalable approach of EG-NAS, while incorporating our bio-inspired operations into the search space. We retain the combination of gradient descent and CMA-ES to explore architecture parameters efficiently. The evolutionary strategy explores multiple directions, avoiding local optima and enhancing the search for robust and diverse architectures.

The compound fitness function integrates cross-entropy loss and cosine similarity to balance performance and diversity during the search, ensuring that the selected architectures perform well across various metrics.

BioNAS-EG provides a scalable solution for neural architecture search that is suited for large and complex datasets since it takes less search time.

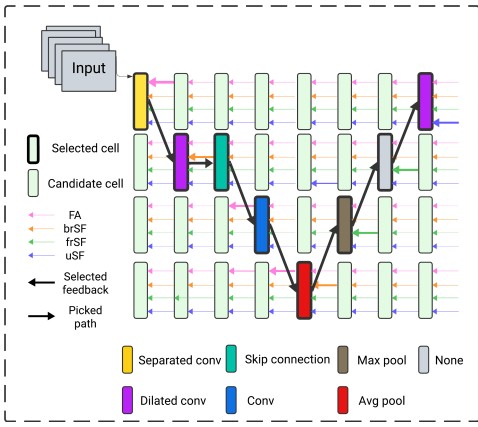

Figure 2: Search process: the supernetwork selects the best candidate operations, which are searched with different learning rules. The backward arrow represents the feedback mechanism (learning rule) chosen.

### 3.4 EXPERIMENTAL SETUP

We benchmark our work on the CIFAR10, CIFAR100 datasets, and the ImageNet16-120 dataset from NasBench201 Dong & Yang (2020). We use a single NVIDIA A100 GPU. We detail here the hyperparameters used for the final architecture, and details about the hyperparameters used for the supernetwork are included in Appendix 4. We also use the package Biotorch to convert the convolutional layers and it uses a Xavier Glorot & Bengio (2010) initialization of the weights to preserve the variance and enhance the training.

#### 3.4.1 CIFAR10 AND CIFAR100

Both CIFAR-10 and CIFAR-100 datasets Krizhevsky (2009) follow the same setup. CIFAR-10 contains 50,000 training images and 10,000 test images across 10 classes, while CIFAR-100 contains 50,000 training images and 10,000 test images across 100 classes.

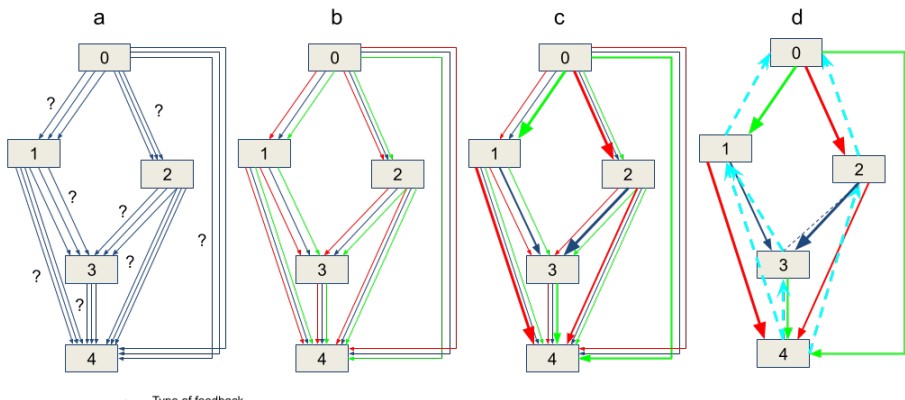

Figure 3: Illustration of BioNAS: Operations on the edges are initially unknown. (b) Continuous relaxation of the search space. (c) Joint optimization of the mixing probabilities and the network weights (d) Inducing the final architecture **as well as the learning rules**. Figure inspired from Karagiannakos (2021)

We train the architectures resulting from the search using the following hyperparameters for both CIFAR-10 and CIFAR-100: batch size of 96, learning rate of 0.025, momentum of 0.9, weight decay of $3 \times 10^{-4}$, for 600 epochs. The architectures are initialized with 36 channels and have 20 layers. We enable the auxiliary tower with a weight of 0.4, apply cutout augmentation with a length of 16, use a drop path probability of 0.2 for BioNAS-DARTS and 0.3 for BioNAS-EG, and apply gradient clipping with a maximum norm of 5. These parameters follow the setup found in Liu et al. (2018) and Cai et al. (2024). For the ResNet20 and ResNet56 architectures, we use the same experimental setup obtained via the hyperparameter search process done by Sanfiz & Akrout (2021b) and reproduce their experiments, and for retraining with the same learning rule, the same setup as the original architecture is applied.

### 3.4.2 IMAGENET16-120

ImageNet16-120 Chrabaszcz et al. (2017) is a subset of ImageNet Deng et al. (2009) with 120 classes, downsampled to 16x16 resolution. This dataset provides a fast benchmark for evaluating the robustness of the searched architectures.

The hyperparameters used to train the architectures on ImageNet16-120 are as follows: batch size of 1024, learning rate of 0.5, momentum of 0.9, weight decay of $3 \times 10^{-5}$, for 250 epochs. The architectures are initialized with 48 channels and have 14 layers. We enable the auxiliary tower with a weight of 0.4, apply label smoothing with a factor of 0.1, use a linear learning rate scheduler, and apply gradient clipping with a maximum norm of 5. No drop path regularization is applied. These settings follow the setup described in Cai et al. (2024).

### 3.5 ADVERSARIAL ATTACKS

In our experiments, we used the following parameters for the adversarial attacks:

- **Fast Gradient Sign Method (FGSM):** We set the maximum perturbation size $\epsilon = 0.35$.

- **Projected Gradient Descent (PGD):** The step size $\alpha$ was set to $2/255$, with a maximum perturbation of $\epsilon = 0.35$ and 10 steps. We also used random initialization.

- **Targeted Projected Gradient Descent (TPGD):** This attack used a step size of $\alpha = 2/255$, a perturbation limit of $\epsilon = 8/255$, and 7 iterations.

| Architecture | Bio-inspired | CIFAR-10 Test Error (%) | CIFAR-100 Test Error (%) |
|---|---|---|---|
| ResNet He et al. (2015) | × | 4.61 | 22.1 |
| ENAS + cutout Pham et al. (2018) | × | 2.89 | - |
| AmoebaNet-A Real et al. (2019) | × | 3.34 | 17.63 |
| NSGA-Net Lu et al. (2019) | × | 2.75 | 20.74 |
| NSGANetV1-A2 Lu et al. (2019) | × | 2.65 | - |
| EPCNAS-C Huang et al. (2022) | × | 3.24 | 18.36 |
| EAEPSO Huang et al. (2023) | × | 2.74 | 16.94 |
| DARTS(1st) Liu et al. (2018) | × | 3.00 | 17.54 |
| DARTS(2nd) Liu et al. (2018) | × | 2.76 | - |
| SNAS (moderate) + cutout Xie et al. | × | 2.85 | 17.55 |
| ProxylessNAS + cutout Cai et al. (2019) | × | 2.02 | - |
| GDAS Dong & Yang (2019b) | × | 2.93 | 18.38 |
| BayesNAS Zhou et al. (2019) | × | 2.81 | - |
| P-DARTS + cutout | × | 2.50 | 17.49 |
| PC-DARTS + cutout Chen et al. (2019) | × | 2.57 | 16.90 |
| DARTS- Xu et al. (2020) | × | 2.59 | 17.51 |
| $\beta$-DARTS Ye et al. (2022) | × | 2.53 | 16.24 |
| DrNAS Chen et al. | × | 2.54 | - |
| DARTS+PT Wang et al. (2023) | × | 2.61 | - |
| EG-NAS Cai et al. (2024) | × | 2.53 | 16.22 |
| ResNet20 FA | ✓ | 32.16 | 75.70 |
| ResNet20 frSF | ✓ | 11.2 | 58.68 |
| ResNet20 brSF | ✓ | 11.02 | 58.37 |
| ResNet20 uSF | ✓ | 10.05 | 56.09 |
| ResNet56 FA | ✓ | 34.88 | - |
| ResNet56 frSF | ✓ | 9.49 | - |
| ResNet56 brSF | ✓ | 8.69 | - |
| ResNet56 uSF | ✓ | 8.2 | - |
| ResNet20 BP | × | 8.63 | - |
| ResNet56 BP | × | 8.3 | - |
| SoftHebb* Moraitis et al. (2022b) | ✓ | 19.60 | - |
| FastHebb Lagani et al. (2022a) | ✓ | 15.0 | - |
| **BioNAS-EG (ours)** | ✓ | 7.16 | 29.76 |
| **BioNAS-DARTS (ours)** | ✓ | **5.14** | 23.52 |

Table 1: Test error rates for CIFAR-10 and CIFAR-100 compared to NAS, human-designed bio-inspired, and BP methods.

**Legend:** ✓ represents bio-inspired architectures; × represents architectures trained using standard backpropagation.

*: Results on CIFAR100 are reproduced from the open-sourced code with the same experimental setup as CIFAR10 and 150 epochs.

The full ImageNet (Deng et al. (2009)) achieves an accuracy of 60%.

- **One-Pixel Attack:** We altered 1 to 6 pixels, using a population size of 10 and 10 optimization steps for the differential evolution.

- **Auto Projected Gradient Descent (APGD):** We used the $L_\infty$ norm with a perturbation size of $\epsilon = 8/255$, 50 steps, and 1 restart.

Each attack was applied in both the untargeted and targeted modes where applicable, to explore the full range of adversarial vulnerabilities in our bio-inspired neural networks.

# 4 RESULTS

Through the experiments reported in Table 1, we show that NAS can effectively enable us to find the best architecture as well as the best bio-inspired learning rule, and we notice that the architecture that DARTS discovers on CIFAR10 performs well when benchmarked on CIFAR100, surpassing the backpropagation based DARTS-V1 from the Nas-Bench-201, ResNet20, and ResNet56 that are both trained with backpropagation. Table 3 shows the accuracy of BioNAS compared to other state-of-the-art NAS methods. Despite having a lower accuracy than BP-trained supernetworks, it is to the best of our knowledge the first attempt to train a supernetwork with mixed bio-inspired learning rules.

We wanted to investigate the role that the learning rules and the architectures take in shaping this searched neural network. In Figure 4, we compare converting a neural network architecture searched independently of the learning rules and converting it to different modes, to a combined search of learning rule and architecture.

We train the architecture searched with BioNAS with backpropagation after converting it to different modes (brSF, frSF, uSF) on CIFAR10, with the same experimental setup used for training the resulting architecture from BioNAS. We notice that mixing the learning rules gives the best performance, which means that both the architecture choice and the learning rule choice are important. In order to underlying learning dynamics, we provide some analysis in subsection A.2, mixing the learning rules yields learning dynamics similar to those observed with L1 and L2 regularisation, yet a theoretical formulation and proof is to be provided.

| Model | Clean Acc | One Pixel attack | | | | | FGSM | PGD | TPGD | APGD |
|---|---|---|---|---|---|---|---|---|---|---|
| | | 1 | 2 | 3 | 4 | 5 | | | | |
| RN56 FA | 65.80 | 47.80 | 40.60 | 38.20 | 38.19 | 38.00 | **61.72** | **63.28** | 67.19 | 33.59 |
| RN56 uSF | 91.60 | 63.80 | 44.00 | 38.40 | 36.40 | 34.40 | 14.60 | 00.00 | 40.62 | 00.00 |
| RN56 frSF | 89.80 | 62.60 | 42.40 | 40.60 | 38.60 | 35.40 | 18.75 | 00.00 | 46.09 | 00.00 |
| RN56 brSF | 89.80 | 65.40 | 45.00 | 42.00 | 40.40 | 38.00 | 73.43 | 00.00 | 50.00 | 00.00 |
| ResNet20 BP | 91.99 | 59.60 | 35.80 | 32.40 | 32.00 | 28.20 | 12.50 | 00.00 | 33.59 | 00.00 |
| **BioNAS-DARTS (ours)** | **94.86** | **90.89** | **89.38** | **87.46** | **86.17** | **84.93** | 61.05 | 60.58 | **67.48** | **66.99** |

Table 2: Accuracy table comparing the robustness of BioNAS to backpropagation for 5 types of adversarial attacks.

## 4.1 DISCUSSION

The key benefit of using mixed learning rules lies in their ability to balance gradient variance across layers as shown in Figure 7 in the Appendix A. The variance in the gradient estimate when using a single rule across the network can be expressed as:

$$\text{Var}[\nabla\mathcal{L}(\theta)] = \frac{1}{n}\sum_{i=1}^{n}(\nabla\mathcal{L}_i(\theta) - \mathbb{E}[\nabla\mathcal{L}_i(\theta)])^2. \tag{8}$$

When using mixed learning rules, the total variance becomes:

$$\text{Var}_{\text{Mixed}}[\nabla\mathcal{L}(\theta)] = \frac{1}{n}\sum_{i=1}^{n} w_i \text{Var}_{\text{Rule}}[\nabla\mathcal{L}_i(\theta)], \tag{9}$$

where $w_i$ is the weight assigned to each learning rule. By combining rules with different characteristics, the overall variance is reduced, resulting in smoother gradient propagation and more stable training. This mixture improves the robustness of the network to variations in gradient magnitude, especially in deeper layers, leading to better generalization.

| Model | CIFAR-10 Test % | CIFAR-100 Test % | ImageNet16-120 Test % | Search Time (GPU-Days) |
|---|---|---|---|---|
| **BioNAS-EG (Bio)** | 81.16 | 70.24 | 23.84 | 0.35 |
| **BioNAS-DARTS (Bio)** | 85.83 | 52.12 | - | 1.37 |
| ResNet He et al. (2015) (BP) | 90.83 | 70.42 | 44.53 | - |
| Random (BP) | 93.70 | 70.65 | 26.28 | - |
| ENAS Pham et al. (2018) (BP) | 53.89 | 13.96 | 14.57 | 0.5 |
| Random-NAS (BP) | 84.07 | 52.31 | 26.28 | - |
| SETN Dong & Yang (2019a) (BP) | 87.64 | 59.05 | 32.52 | - |
| GDAS Dong & Yang (2019b) (BP) | 93.23 | 24.20 | 41.02 | 0.2 |
| DSNAS Hu et al. (2020) (BP) | 93.08 | 31.01 | 41.07 | 1.5 |
| DARTS-V1 Liu et al. (2018) (BP) | 54.30 | 15.61 | 16.32 | 0.4 |
| DARTS-V2 Liu et al. (2018) (BP) | 54.30 | 15.61 | 16.32 | 1.0 |
| PC-DARTS (BP) Xu et al. (2020) | 93.41 | 67.48 | 41.31 | 0.1 |
| iDARTS (BP) Wang et al. (2023) | 93.58 | 70.83 | 40.8 | - |
| DARTS Chu et al. (2020) (BP) | 93.80 | 71.53 | 45.12 | 0.4 |
| EGNAS Cai et al. (2024) (BP) | 93.56 | 70.78 | 46.13 | 0.1 |

Table 3: Validation accuracies and search times for the **supernetwork** on CIFAR-10, CIFAR-100, and ImageNet16-120 datasets. BioNAS's supernetwork achieves a validation accuracy of 23.84% in the search phase and an accuracy of 43.42% when the architecture is trained separately

| Model | Top-1 accuracy % |
|---|---|
| BioNAS-EG-brSF | 83.17 |
| BioNAS-EG-frSF | 90.58 |
| BioNAS-EG-uSF | 86.54 |
| BioNAS-EG (ours) | **92.84** |
| BioNAS-DARTS-brSF | 83.17 |
| BioNAS-DARTS-frSF | 90.58 |
| BioNAS-DARTS-uSF | 86.54 |
| BioNAS-DARTS (ours) | **94.86** |

Figure 4: Comparing neural architecture search to neural architecture search with learning rules.

## 5 CONCLUSION

In this paper, we show that a good choice of the learning rule used for each layer in a network and the operation can result in a bio-inspired neural network with a good accuracy competing with backpropagation-trained neural networks, achieving state-of-the-art bio-inspired neural network accuracy. The mixed-rule approach provides advantages over using a single learning rule throughout the network by reducing gradient variance, facilitating better exploration of the loss landscape, and enhancing adversarial robustness. This theoretical framework aligns with the empirical results observed in our experiments, where architectures searched using NAS with mixed learning rules outperform those using a uniform rule across all layers. Further investigation should be done on the learning dynamics in this searched architecture and the representations learned.

## 6 ETHICS STATEMENT

We acknowledge the significant computational resources required for the NAS experiments on GPUs. However, we believe this work contributes to the development of more robust, bio-inspired neural networks, which can lead to more secure and efficient AI systems in the long run.

## 7 REPRODUCIBILITY STATEMENT

To ensure the reproducibility of our work, we have made the experimental code publicly available along with detailed instructions for setup and execution. This includes all necessary configurations and steps to replicate the results presented in this paper.

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

## A  APPENDIX

### A.1  SUPERNETWORK HYPERPARAMETERS

Table 4 summarizes the hyperparameters used for the supernetworks training for CIFAR10 and CIFAR100. When it comes to ImageNet16-120, an initial learning rate of 0.5 is used, and a batch size of 512.

| Hyper-parameter | Value |
| --- | --- |
| Optimizer | SGD |
| Initial LR | 0.1 |
| Nesterov | Yes |
| Ending LR | 0.0 |
| Momentum | 0.9 |
| LR Schedule | (Not mentioned) |
| Weight Decay | 0.0003 |
| Epochs | 50 |
| Batch Size | 256 |
| Initial Channels | 16 |
| V (Nodes per Cell) | 4 |
| N (Cells) | 5 |
| Random Flip | p=0.5 |
| Random Crop | Yes |
| Normalization | Yes |

Table 4: Hyperparameters used for training the supernetwork on CIFAR10 and CIFAR100.

### A.2  WEIGHT DISTRIBUTION ANALYSIS

To understand how the learning happens, we plot the weight distribution of our model as shown in Figure 5, the distribution is a little bit steeper than a Gaussian which is an effect observed usually with regularization.

We compare the weights of the models with the highest accuracies, we train the resulting BioNAS-EG architecture with the same learning rule from end to end, as shown in figure 6, the weights of BioNAS-EG is steeper with more weights around 0. Note that we drop the weights that are less than 2 and more than 2 since their number is negligible and invisible on the plot.

Another thing we noticed is that for training with the same learning rule, the variance is much smaller than with BioNAS-EG as illustrated in figure 7.

### A.3  THEORETICAL JUSTIFICATION OF MIXED LEARNING RULES

To demonstrate why a neural architecture searched with NAS using mixed learning rules outperforms architectures that use the same learning rule across all layers, we analyze the impact of combining various learning rules on gradient propagation, variance reduction, and optimization efficiency.

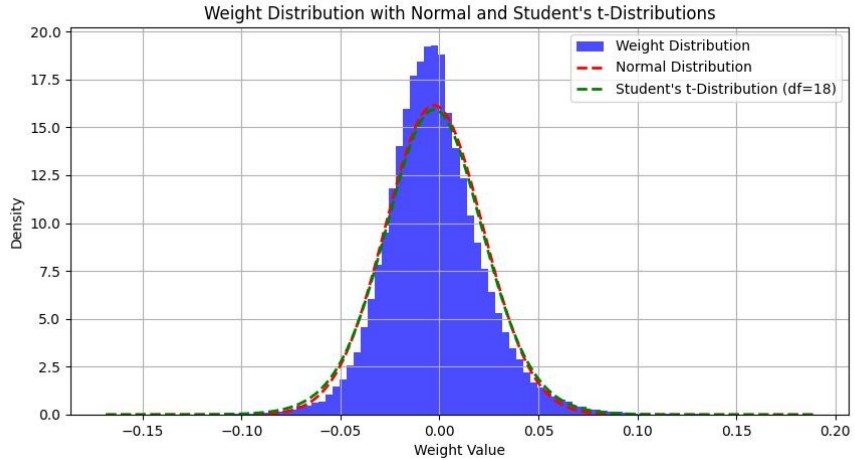

Figure 5: Weight distribution of BioNAS-EG compared to a gaussian and and a student distribution with 10 degrees of freedom.

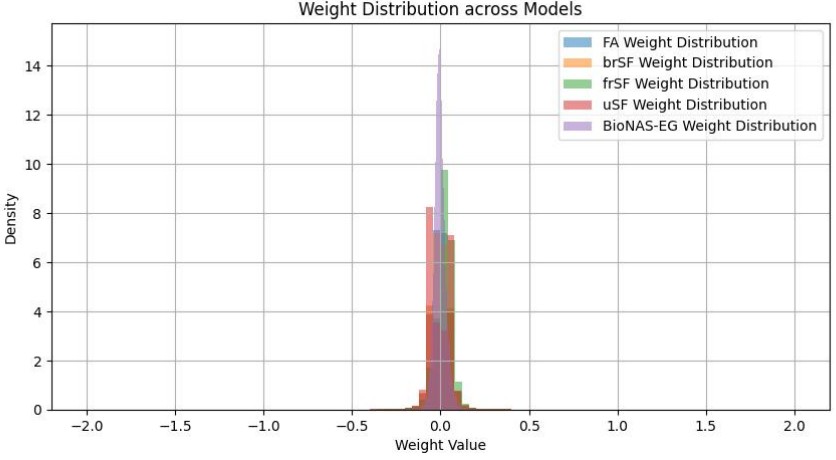

Figure 6: Weight distribution comparison between BioNAS-EG vs training it with the same learning rule (FA, frSF, brSF, uSF) from end to end.

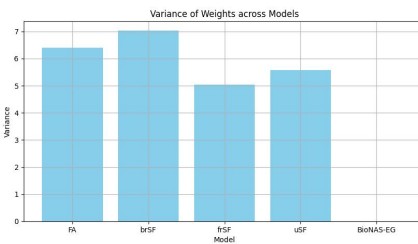 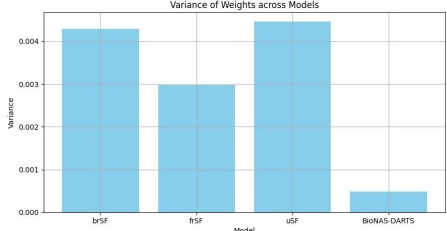

Figure 7: Comparison of weight variance and performance between BioNAS-EG (left) and BioNAS-DARTS (right) models (with mixed learning rules) versus training the same resulting architecture with the same learning rule from end-to-end.

### A.3.1 GRADIENT PROPAGATION AND LEARNING DYNAMICS

Consider a neural network consisting of multiple layers, each trained with different feedback learning rules. The general weight update in a layer $i$ can be written as:

$$\delta W_i = -(B_{i+1}\delta \mathbf{z}_{i+1})\phi'(\mathbf{z}_i)\mathbf{y}_{i-1},$$

where $B_{i+1}$ is the feedback matrix and $\delta \mathbf{z}_{i+1}$ is the gradient of the loss with respect to the input of layer $i + 1$.

When a network uses a single learning rule throughout, this feedback matrix $B_{i+1}$ is updated uniformly across all layers. However, when different learning rules are applied in different layers, each rule adapts the feedback matrix differently, allowing for a broader range of gradient dynamics across the network. This diversity in updates helps mitigate issues like gradient vanishing or explosion, promoting better gradient flow.

### A.3.2 EXPLORATION AND OPTIMIZATION EFFICIENCY

Each learning rule contributes differently to the exploration of the loss landscape. The variation across layers helps the network strike a balance between exploration and exploitation during optimization. For example, some rules might encourage the network to explore different parts of the loss surface by generating different gradient directions, while other rules help converge efficiently by stabilizing the updates.

The overall gradient update at time $t$ for a mixed-rule network is:

$$\Delta \theta_t^{\text{Mixed}} = \sum_k w_k \Delta \theta_t^k,$$

where $\Delta \theta_t^k$ is the gradient update under rule $k$ and $w_k$ is the weight assigned to that rule. By combining these updates, the mixed-rule network benefits from diverse gradient dynamics that prevent overfitting and improve convergence.

### A.3.3 ADVERSARIAL ROBUSTNESS

Using mixed learning rules enhances adversarial robustness by disrupting the consistency in gradient signals across layers. In adversarial attacks, the goal is to align perturbations with the network's gradient, but with different learning rules in different layers, the adversary faces a moving target. This makes it harder to craft perturbations that generalize across the network, improving resistance to adversarial examples.

## A.4 NEURAL ARCHITECTURE SEARCH

### A.4.1 DARTS

In DARTS, a continuous relaxation of the architecture search space is employed, enabling the use of gradient-based optimization to search for the optimal architecture. Specifically, each intermediate node is computed by applying a softmax function over a mixture of candidate operations:

$$o^{(i,j)}(x^{(i)}) = \sum_{o \in O} \frac{\exp(\alpha_o^{(i,j)})}{\sum_{o' \in O} \exp(\alpha_{o'}^{(i,j)})} o(x^{(i)}),$$

where $i < j$, the set of candidate operations is represented by $O$, and $\alpha_o^{(i,j)}$ represents the mixing weight for operation $o^{(i,j)}$ in the supernetwork. During the search process, DARTS optimizes both the network weights $\omega$ and the architecture parameters $\alpha$ simultaneously using a bi-level optimization framework. This framework is defined as:

$$\min_{\alpha} F(\omega^*(\alpha), \alpha) = L_{\text{val}}(\omega^*(\alpha), \alpha)$$
$$\text{subject to } \omega^*(\alpha) = \arg\min_{\omega} L_{\text{train}}(\omega, \alpha),$$

where $\omega^*(\alpha)$ denotes the optimal network weights for a given architecture. Both $\alpha$ and $\omega$ are updated through gradient descent. After the search phase concludes, the final architecture is derived by selecting the operation with the highest architectural parameter $\alpha$ on each edge:

$$o^{(i,j)} = \arg\max_{o \in O} \alpha_o^{(i,j)}.$$

### A.4.2 EG-NAS

In traditional NAS approaches, gradient-based methods typically calculate gradients and use them to guide the search for optimal architectures. However, these methods are often limited by the gradient's inherent directionality, leading to a potential risk of getting trapped in local minima. EG-NAS addresses this issue by combining Covariance Matrix Adaptive Evolutionary Strategy (CMA-ES) with gradient-based optimization. CMA-ES is a powerful evolutionary algorithm known for its global search capabilities and efficient convergence in black-box optimization problems Hansen (2016); Loshchilov & Hutter (2016).

The EG-NAS algorithm begins by sampling $N$ architectures, $\{x_n\}_{n=1}^N$, from a Gaussian distribution with a mean vector $\alpha$ and covariance matrix $I$, such that:

$$x_n = \alpha + \sigma y, \quad y \sim \mathcal{N}(0, I), \quad n = 1, 2, \ldots, N.$$

Each sampled architecture $x_n$ initializes a CMA-ES search, where $x_n$ serves as the mean vector $m_0$ of the $n$-th search. The initial population of the $n$-th evolutionary search is sampled as:

$$z_i^t = m^t + \sigma y_i, \quad y_i \sim \mathcal{N}(0, C_t),$$

where $t$ represents the iteration index, starting from $t = 0$, with $C_0 = I$ and $\sigma$ as the step size. During each iteration, the mean vector $m^{t+1}$ and covariance matrix $C_{t+1}$ are updated using the following equations:

$$m^{t+1} = \sum_{i=1}^{\lfloor \lambda/2 \rfloor} \beta_i z_i^t,$$

$$C_{t+1} = (1 - c_1 - c_{\lfloor \lambda/2 \rfloor})C_t + c_1(pp^T) + c_{\lfloor \lambda/2 \rfloor} \sum_{i=1}^{\lfloor \lambda/2 \rfloor} \beta_i(z_i^t - m^{t+1})(z_i^t - m^{t+1})^T,$$

where $\beta_i$ represents the fitness weight for each individual, and $p$ is the evolutionary path.

Each individual is evaluated by a compound fitness function, incorporating both the cross-entropy loss ($L_1$) and cosine similarity ($L_2$):

$$f(\alpha^t, z_i^{t+1}) = \begin{cases} \zeta L_1 - \eta L_2 & \text{if } \text{Acc}(\alpha^t) > \text{Acc}(z_i^{t+1}) \\ \zeta L_1 + \eta L_2 & \text{else}, \end{cases}$$

where $\zeta$ and $\eta$ are weight coefficients, and $L_2$ ensures diversity among architectures by calculating the cosine similarity between $\alpha^t$ and $z_i^{t+1}$. This compound fitness function helps avoid premature convergence and ensures exploration of diverse architectures.

Finally, EG-NAS integrates the evolutionary strategy with gradient-based optimization. The network weights $\omega$ are updated via gradient descent, while the architecture parameters $\alpha$ are updated using the evolutionary strategy. The update rule for $\alpha$ is:

$$\alpha^t = \alpha^{t-1} + \xi s^t,$$

where $s^t$ represents the search direction based on fitness values, and $\xi$ is the step size. The best-performing architecture across all evolutionary steps is selected as the final output, ensuring a more diverse and effective architecture search than traditional gradient-based methods.

