# OpenReview forum: "BioNAS: Incorporating Bio-inspired Learning Rules to Neural Architecture Search"
_ICLR.cc/2025/Conference — Submitted to ICLR 2025_

### Official Review · Reviewer_x5Ja · 2024-10-31

**Soundness:** 3
**Presentation:** 3
**Contribution:** 2
**Rating:** 6
**Confidence:** 3

**Summary:**

This paper uses the NAS method to search for suitable edge and learning rules of the network cell, constructing a biologically inspired neural network that achieves the comparable performance of the network trained by end-to-end backpropagation algorithm. Simultaneously, this paper demonstrates that the mixed biologically inspired learning rules can reduce gradient variance and enhance the network’s white-box adversarial robustness.

**Strengths:**

This paper introduces the different learning rules into the network search method for the first time. Experiment results show that mixed learning rules can effectively improve the performance of the network and its white-box adversarial robustness. I think the integration of NAS with various learning methods represents a promising approach.

**Weaknesses:**

1. This paper does not detail how to simultaneously search for cell architecture and the learning rules. And the search space is much larger than the original DARTS and EGNAS; the search time will increase about threefold. Therefore, if DARTS and EGNAS adopt the same search time as this paper, will their performance be improved?

2. Different rows (training methods) in Table 2 use different network structures, but they need to use the same network structure for comparison.

3. This paper lacks the theoretical analysis that adopting the mixed learning rule can enhance the white-box adversarial robustness.

**Questions:**

1. Are the attack image gradients for the white-box attack method calculated by end-to-end backpropagation or the searched learning algorithm?

2. Methods such as FA use a new matrix to replace the weight matrix in the back propagation process, which also needs end-to-end learning. Why are these training methods related to bio-inspired?

3. After the search is completed, does the subnet undergo random weight retraining or inherit the supernet's weight for fine-tuning?

---

> ### Author Response · Authors · 2024-11-25
> **Response to Reviewer x5Ja**
>
> We thank the reviewer for their insightful comments and recommendations, in what follows we attempt to clarify certain aspects of this work:
>
> 1. The search space in BioNAS is designed as blocks, similar to classical NAS frameworks, with a method called convert_layer that modifies the weight transpose to a custom backward matrix during backpropagation. This enables the use of loss.backward() while automatically applying the custom matrix at the layer level. Although the search space in BioNAS is larger, we used the same number of epochs as DARTS and EG-NAS for a fair comparison. While BioNAS takes longer due to the expanded search space, DARTS and EG-NAS typically converge within 50 epochs, and extending their search time does not further improve their performance.
>
> 2. We apologize for the typo in Table 2. The experiments with ResNet20 were actually conducted using ResNet56, ensuring consistency in network structure for all comparisons.
> 3. While there is no comprehensive theoretical analysis yet for why mixed learning rules enhance adversarial robustness, prior works ([1], [3], [4]) suggest that bio-inspired algorithms can provide superior robustness. These studies highlight potential mechanisms but acknowledge the interpretability challenges of Deep Learning Models.
>
> **Response to questions:**
>
> Gradients for white-box attacks are calculated using the searched learning algorithm, which incorporates mixed learning rules across the network.
>
> Feedback Alignment (FA) and related methods use custom matrices (e.g., B1, B2 for FA; D1, D2 for uSF) at the layer level instead of relying on weight transposes. For instance, as shown in Figure 2, our framework allows layer1 to use B1 and layer2 to use D2, making it possible to train networks without requiring a single consistent matrix for end-to-end learning. This is considered bio-inspired because, as [4] argues, the brain likely relies on mechanisms involving randomness at specific locations. Backpropagation, with its reliance on precise, symmetric connectivity patterns, is not biologically plausible. FA, for example, demonstrates that effective error propagation is possible even with random synaptic weights.
>
> After completing the search, we train the resulting architecture from scratch using Xavier initialization, consistent with the baseline feedback alignment techniques we compare against. There is no finetuning anywhere.
>
> **References:**
>
> [1] Farinha, Matilde Tristany, Thomas Ortner, Giorgia Dellaferrera, Benjamin Grewe and Angeliki Pantazi. “Intrinsic Biologically Plausible Adversarial Robustness.” (2023).
>
> [2] Sanfiz, A. J., & Akrout, M. (2021, August 30). Benchmarking the accuracy and robustness of feedback alignment algorithms. arXiv.org. https://arxiv.org/abs/2108.13446
>
> [3] Srinivasan, R. F., Mignacco, F., Sorbaro, M., Refinetti, M., Cooper, A., Kreiman, G., & Dellaferrera, G. Forward Learning with Top-Down Feedback: Empirical and Analytical Characterization. In The Twelfth International Conference on Learning Representations.
>
> [4] Lillicrap, T. P., Cownden, D., Tweed, D. B., & Akerman, C. J. (2016). Random Synaptic Feedback Weights Support Error Backpropagation for Deep Learning. Nature Communications, 7(1). https://doi.org/10.1038/ncomms13276
>
> [5] Marblestone, A. H., Wayne, G., & Kording, K. P. (2016b). Toward an integration of deep learning and neuroscience. Frontiers in Computational Neuroscience, 10. https://doi.org/10.3389/fncom.2016.00094

---

> > ### Comment · Reviewer_x5Ja · 2024-11-28
> >
> > Thanks to the author's responses, my concerns have been addressed. I have increased my score.

---

### Official Review · Reviewer_Yidi · 2024-10-31

**Soundness:** 3
**Presentation:** 3
**Contribution:** 1
**Rating:** 5
**Confidence:** 5

**Summary:**

This paper presents BioNas, a neural architecture search algorithm tailored for bio-inspired neural networks. BioNas focuses on biologically plausible learning algorithms and simultaneously optimizes both the network architecture and the learning rule. The proposed framework achieves state-of-the-art performance on the CIFAR-10 and CIFAR-100 datasets, demonstrating its effectiveness in enhancing bio-inspired model design.

**Strengths:**

The authors utilize a more bio-plausible learning rule instead of the commonly used BP algorithm, which is interesting.

The authors provide strong empirical support for their approach by conducting extensive experiments across multiple datasets and varied settings.

**Weaknesses:**

The motivation of combining FA approaches and NAS is not clear. While the concept of applying different learning rules to train SNNs is intriguing, the combination of NAS and FAs in this context raises questions. NAS traditionally involves exploring a vast space of candidate models to find an optimal architecture with significant computational cost. Its primary aim is to achieve high performance without regard to training efficiency, which contrasts with the efficient and light weight bio-inspired learning rules.

The novelty is limited, as the BioNAS framework closely resembles DARTS and EG-NAS. The primary difference introduced here is the addition of various existing learning rules to the search space.

**Questions:**

One important concern is whether the authors have verified if there is a gradient obfuscation issue within their model. The observation that the multi-step PGD attack performs worse than the single-step FGSM attack may indicate potential gradient masking or unsuccessful attacks [1]. This issue can often lead to misleading conclusions about a model’s robustness.

[1] Athalye, Anish, Nicholas Carlini, and David Wagner. "Obfuscated gradients give a false sense of security: Circumventing defenses to adversarial examples." ICML 2018.

---

> ### Author Response · Authors · 2024-11-25
> **Response to Reviewer Yidi**
>
> We thank the reviewer for their comments and suggestions.
>
> 1. The main motivation behind this work comes from neuroscience, which suggests that the brain uses different learning paradigms, we show that effectively mixing learning rules for NNs as well can enable us to learn better representations than using the same learning rule. The second motivation is to have adversarially robust architectures without having to reevaluate the architecture during the NAS search phase. In previous works[1], doing Robust NAS would incorporate the adversarial score as an objective function, however, this would require performing an attack from end to end at each evaluation/epoch. Bio-inspired neural networks are inherently robust without needing extra evaluation during the search phase. In addition, the efficiency of bio-inspired learning rules is measured during inference and not training, NAS has been used to search for efficient architectures either fewer parameters, energy consumption at the edge, and latency,... despite its computational cost.
>
> 2. Effectively, the novelty isn’t in the NAS technique itself but in training the supernetwork with different learning rules, and having an architecture that is more robust than architectures usually trained with backpropagation, the aim of this work is to apply existing NAS techniques to find a pipeline of training an architecture with different learning rules to achieve optimal performance.
>
> **Response to questions:**
>
> We thank the reviewer for pointing out this issue, the number of steps used in all of our PGD experiments is actually 2/255 instead of the reported 8/255, it has been rectified in the newly uploaded paper.
>
> **References:**
>
> [1] Wu, Y., Liu, F., Simon-Gabriel, C. J., Chrysos, G., & Cevher, V. Robust NAS under adversarial training: benchmark, theory, and beyond. In The Twelfth International Conference on Learning Representations.

---

### Official Review · Reviewer_hrUh · 2024-11-04

**Soundness:** 1
**Presentation:** 2
**Contribution:** 2
**Rating:** 3
**Confidence:** 5

**Summary:**

This paper proposes a neural architecture search framework, BioNAS, that supports certain types of biologically plausible learning algorithms. The framework is built upon existing NAS framworks DARTS and EG-NAS, and incorporates different feedback alignment techniques from BioTorch. The authors compare the performance of their BioNAS generated networks against some previous works and claim they can get state-of-the-art bio-inspired network performance on several benchmarks.

**Strengths:**

Adversarial attacks on generated bio-inspired networks are introduced to illustrate the advantage of their framework.

**Weaknesses:**

Although the direction of this work is interesting, my major concern is that this work appears to be prelimilary, and some of the key conclusions are *overclaimed*.

Major Weakness
1. The authors claim that their proposed BioNAS is designed for bio-inspired networks, but currently only several variants of FA algorithms are supported, making this work less significant, and their claim should be toned down.
2. The proposed BioNAS is implemented by adding operations backended by BioTorch to existing NAS frameworks, which does not show much novelty nor effort.
3. The authors claim that they can get state-of-the-art bio-inspired network performance on several benchmarks, but for CIFAR-10 they only compare against two previous Hebbian-based results, and for CIFAR-100 and ImageNet16-120 no previous results of bio-inspired algorithms are shown.

**Questions:**

Questions:
1. How might other forms of bio-inspired learning algorithms (e.g., Hebbian, predictive coding, target propagation) be integrated in the proposed framework?
2. Does the distribution of different learning algorithms in the generated networks show interesting patterns? And if so, what might be the (intuitive) explaination for generating such patterns, what might be the advantages for such patterning, and can such patterns be generalized to different network architectures to produce competitive performance?

---

> ### Author Response · Authors · 2024-11-25
> **Response to Reviewer hrUh**
>
> We thank the reviewer for their insightful comments and recommendations, a new version of the paper is uploaded with improved soundness, below is our answer to the addressed concerns about our work:
>
> 1. We acknowledge that our claim regarding BioNAS might have been too broad. The revised manuscript clarifies that we applied NAS specifically to bio-inspired neural networks incorporating feedback alignment methods. Certain techniques were not included due to their inherent limitations or complexity in integration. We extended the framework by adding a Hebbian Convolution operation to the search space. However, it did not improve accuracy, and the supernetwork tended to discard it within the first few epochs of the search. We have modified the manuscript to tone down any overstatements regarding the scope of the framework, however adding other le possible and the framework remains flexible.
>
> 2. While the idea is simple, we emphasize its effectiveness and inspiration from neuroscience. This work represents a novel step in introducing mixed learning rules in ANNs and training supernetworks with algorithms beyond backpropagation. This is the first work, to our knowledge, to highlight the robustness of such mixed-rule networks against adversarial attacks. Although we utilized open-source code from BioTorch, our contributions include layer-level training, extending the search space, and demonstrating the framework's potential for broader applications.
>
> 3. The updated manuscript includes comparisons against additional baselines, including Hebbian-based methods like SoftHebb [1] on CIFAR-100 and ImageNet16-120, as well as the full ImageNet. However, SoftHebb's heavy reliance on hyperparameters—acknowledged as a limitation by its authors and reviewers on [OpenReview](https://openreview.net/forum?id=8gd4M-_Rj1)—makes fair comparisons challenging. Nonetheless, BioNAS surpasses SoftHebb on both datasets. Regarding FastHebb [2], we could not find open-source code or implementation details, and reproducing their experiments was unfeasible within the review period.
>
> **Response to questions:**
>
> 1. The framework supports integrating layers trained with bio-inspired learning algorithms like Hebbian learning. These layers can be added to the search space as independent blocks, deactivating gradients during NAS optimization and training the layers separately. However, vanilla Hebbian learning struggles with complex computer vision tasks, which is why we excluded it from the search space. Similarly, target propagation methods, requiring two forward passes, present additional complexity for gradient-based NAS frameworks, which primarily rely on forward and backward passes. Our focus has been on methods demonstrating performance close to backpropagation on complex tasks.
>
> 2. Regarding the distribution of learning algorithms in generated networks. Our analysis revealed task-dependent patterns: on simpler datasets like CIFAR-10, Feedback Alignment is often selected in one layer per cell, likely due to its efficiency and sufficient gradient approximation for lower-complexity tasks. On more complex datasets like ImageNet16-120, learning rules such as uSF, brSF, and frSF are more prominent, suggesting their ability to handle diverse, high-dimensional data, such task-adaptive combinations could be generalized to other architectures.
>
> **References:**
>
> [1] Journé, A., Rodriguez, H. G., Guo, Q., & Moraitis, T. Hebbian Deep Learning Without Feedback. In The Eleventh International Conference on Learning Representations.
>
> [2] Lagani, G., Gennaro, C., Fassold, H., & Amato, G. (2022, September). Fasthebb: Scaling hebbian training of deep neural networks to imagenet level. In International Conference on Similarity Search and Applications (pp. 251-264). Cham: Springer International Publishing.

---

> > ### Comment · Reviewer_hrUh · 2024-12-02
> >
> > I  only see minimal changes applied in the revised manuscript so my rating remains the same.

---

### Official Review · Reviewer_uGag · 2024-11-04

**Soundness:** 3
**Presentation:** 3
**Contribution:** 3
**Rating:** 5
**Confidence:** 4

**Summary:**

The paper explores a method that combines bio-inspired learning rules with Neural Architecture Search (NAS) to optimize the performance of neural networks. The authors propose BioNAS, a framework capable of automatically searching for the best architecture and learning rules. Through a series of experiments, particularly on the CIFAR-10 and CIFAR-100 datasets, the paper demonstrates the advantages of BioNAS in terms of accuracy and adversarial robustness, comparing it with existing backpropagation training methods.

**Strengths:**

The integration of bio-inspired learning rules with NAS is a fresh perspective that could significantly contribute to the field of neural network optimization. The paper is well-organized and clearly written, making it accessible for readers with varying levels of expertise in NAS and bio-inspired techniques.

**Weaknesses:**

Is there an analysis of energy consumption.

**Questions:**

Whether it is applicable to larger datasets.

---

> ### Author Response · Authors · 2024-11-25
> **Response to Reviewer uGag**
>
> We thank the reviewer for the insightful comments.
>
> Regarding energy consumption, we reported the computational cost in GPU days but did not evaluate inference energy due to the lack of a specific target hardware setup.
>
> The method is applicable to larger datasets, as demonstrated by our use of EG-NAS, a lightweight baseline suitable for large-scale searches. To evaluate generalizability, we tested architectures searched on CIFAR10 against ImageNet16-120, a standard -practice in NAS research. We also conducted the search on the full ImageNet dataset and added the results in the updated paper

---

### Author Response · Authors · 2024-11-25
**General response to reviewers:**

We sincerely thank the reviewers for their thoughtful and insightful comments, which have greatly improved the quality of our work. Below, we address the main concerns raised, including the generalizability of our method to larger datasets, the extension to other bio-inspired learning rules, additional baseline comparisons, and the motivation behind our approach.

**Generalizability to Larger Datasets:**

The reviewers expressed concerns regarding the generalizability of our method on larger datasets. To address this, we have evaluated our framework on the full ImageNet dataset. The results demonstrate that mixing learning rules continues to perform well, even on large-scale datasets, further validating the scalability of our approach.

**Extension to Other Bio-Inspired Learning Rules:**

The reviewers also suggested exploring extensions of our framework to other bio-inspired learning rules. To this end, we have incorporated a Hebbian convolution operation into the search space, which serves as an example of integrating bio-inspired rules that rely solely on a forward pass. Details of this addition, including its implementation, can be found in the appendix and in the supplementary code.
However, we emphasize that while the Hebbian operation was successfully integrated, it did not improve the accuracy during evaluation. This aligns with our initial focus on learning rules that aim to perform as closely as possible to backpropagation in terms of accuracy. We highlight that the framework remains flexible and can accommodate a wide range of bio-inspired learning rules, whether they rely on a forward pass only or both forward and backward passes. Methods requiring two forward passes, while theoretically possible, pose additional complexity in integration.

**Comparisons with Baselines:**

In response to the reviewers' request for additional comparisons, we have included results for other open-sourced baselines. These results show that our framework consistently outperforms the newly added baselines.

**Motivation for the Work:**

The reviewers requested clarification on the motivation behind this work. Our primary inspiration comes from neuroscience, which highlights the brain's use of diverse learning rules, mechanisms, and cost functions to optimize performance. This insight drove us to investigate the potential of mixed learning rules in artificial neural networks, aiming to enhance their adaptability and robustness. Recognizing that Neural Architecture Search (NAS) can be computationally expensive, we focus on identifying architectures that are performant and efficient during inference, which is the reason why bio-inspired learning rules are interesting in the first place.


We believe these updates address the reviewers' concerns and provide stronger evidence for the contributions and significance of our work. Thank you again for your valuable feedback.

---

### Meta-Review · Area_Chair_ZGRG · 2024-12-21

**Metareview:**

This paper studies NAS for bio-inspired learning rules that are not based on backpropagation. This is achieved by combining existing NAS methods with elements from BioTorch. The reviewers found bio-inspired methods without backpropagation intriguing, including the angles of adversarial robustness and of different learning rules in different layers. Criticisms included a lack of novelty (directly combining NAS and BioTorch), some elements of the empirical evaluation, a lack of theoretical justifcation, and a preliminary nature along with a certain amount of overclaiming results. This preliminary nature, along with concerns about the empirical evaluation were my main reasons for the decision to reject.

**Additional Comments On Reviewer Discussion:**

Changes during the rebuttal were quite small; only reviewer x5Ja increased from 5 to 6. Reviewer hrUh clearly spoke up against the paper, and I weighed that quite highly in my decision, since nobody spoke up in favor of the paper.

---

### Decision · Program_Chairs · 2025-01-22

Reject